# Potential Response Patterns of Endogenous Hormones in Cliff Species *Opisthopappus taihangensis* and *Opisthopappus longilobus* under Salt Stress

**DOI:** 10.3390/plants13040557

**Published:** 2024-02-19

**Authors:** Yimeng Zhang, Yuexin Shen, Mian Han, Yu Su, Xiaolong Feng, Ting Gao, Xiaojuan Zhou, Qi Wu, Genlou Sun, Yiling Wang

**Affiliations:** 1School of Life Sciences, Shanxi Normal University, Taiyuan 030031, China; 222112063@sxnu.edu.cn (Y.Z.); 222112064@sxnu.edu.cn (Y.S.);; 2Department of Botany, Saint Mary’s University, Halifax, NS B3H 3C3, Canada

**Keywords:** *Opisthopappus taihangensis*, *Opisthopappus longilobus*, salt stress, endogenous hormone

## Abstract

When plants are exposed to salt stress, endogenous hormones are essential for their responses through biosynthesis and signal transduction pathways. However, the roles of endogenous hormones in two cliff species (*Opisthopappus taihangensis* and *Opisthopappus longilobus* (*Opisthopappus* genus)) in the Taihang Mountains under salt stress have not been investigated to date. Following different time treatments under 500 mM salt concentrations, 239 differentially expressed gene (DEG)-related endogenous hormones were identified that exhibited four change trends, which in Profile 47 were upregulated in both species. The C-DEG genes of AUX, GA, JA, BR, ETH, and ABA endogenous hormones were significantly enriched in *Opisthopappus taihangensis* (*O. taihangensis*) and *Opisthopappus longilobus* (*O. longilobus*). During the responsive process, mainly AUX, GA, and JA biosynthesis and signal transduction were triggered in the two species. Subsequently, crosstalk further influenced BR, EHT, ABA, and MAPK signal transduction pathways to improve the salt resistance of the two species. Within the protein–protein interactions (PPI), seven proteins exhibited the highest interactions, which primarily involved two downregulated genes (*SAUR* and *GA3ox*) and eight upregulated genes (*ACX*, *MFP2*, *JAZ*, *BRI1*, *BAK1*, *ETR*, *EIN2*, and *SNRK2*) of the above pathways. The more upregulated expression of *ZEP* (in the ABA biosynthesis pathway), *DELLA* (in the GA signaling pathway), *ABF* (in the ABA signaling pathway), and *ERF1* (in the ETH signaling pathway) in *O. taihangensis* revealed that it had a relatively higher salt resistance than *O. longilobus*. This revealed that the responsive patterns to salt stress between the two species had both similarities and differences. The results of this investigation shed light on the potential adaptive mechanisms of *O. taihangensis* and *O. longilobus* under cliff environments, while laying a foundation for the study of other cliff species in the Taihang Mountains.

## 1. Introduction

With the rapidly changing climate, soil salinization has emerged as a global scale environmental issue [1]. More than 900 million hectares of land are currently affected by excessive salt worldwide, a problem exacerbated by global warming and anthropogenic activities [2], and further aggravated through the deterioration of the natural environment overall [3].

As is well known, salinity is a common abiotic stress for organisms. The impacts of salinity are increasing rapidly on a global scale, which severely limits plant growth, productivity, and geographical distribution. To facilitate their survival and development, plants typically retain an array of mechanisms to mitigate salt stress, which encompass hormonal stimulation, ion exchange, antioxidant enzymes, and the activation of signaling cascades on their metabolic and genetic frontiers [4].

Among these specific processes, endogenous hormones, which are substances that serve as signaling molecules in response to environmental stress, play key roles in the salt tolerance of plants [5]. During stress responses, hormones including auxin (AUX), abscisic acid (ABA), cytokinin (CK), ethylene (ETH), gibberellin (GA), salicylic acid (SA), jasmonic acid (JA), and brassinolide (BR) take on various functions at different growth stages under diverse conditions [6]. These endogenous hormones regulate the adaptability of plant growth by adjusting saline signals, whereafter plants develop their defense strategies by directing the synthesis, signal transduction, and metabolism of various hormones.

For AUX, it plays pivotal roles in various biological processes, including apical dominance, embryonic development, adventitious root formation of lateral roots, and vascular tissue differentiation [7]. Upon sensing AUX, the receptors initiate the formation of SKP1, Cullin, and F-box (SCF) complexes. These complexes bind to AUX/IAA inhibitors, which leads to ubiquitination and subsequent proteasome-mediated degradation of AUX/IAA. AUX/IAA degradation results in the release of AUX response factors (ARF) and the activation of AUX-induced gene expression. In *Arabidopsis*, mutants lacking AUX receptors exhibit heightened sensitivity to salt stress, accompanied by the downregulation of AUX receptor genes (*TIR1* and *AFB2*). This suggests that *Arabidopsis* mitigates its growth rate to improve salt tolerance by sustaining a diminished AUX signal response [8,9].

GA can promote stem elongation and regulates the development of meristems, biotic, and abiotic stresses [10,11]. It associates with the GOD1 receptor to induce conformational changes. Subsequently, they bind with the DELLA protein, resulting in the formation of a GA-GID1-DELLA complex, which facilitates the degradation of the DELLA protein through the 26S proteasome, thereby activating the downstream response genes [12,13]. Reduced GA levels induce slower growth and assists with improving the stress resistance of plants [14].

ABA synthesis primarily occurs in vascular tissues, and subsequently translocated to guard cells where it modulates responses to osmotic and salt stress, mainly by regulating plant stomata [15]. As a primary mediator, ABA enhances plant survival under salt stress through the activation of plasma membrane-bound channels or by interacting with Ca^2+^ [16]. PYR/PYL is an ABA signaling complex receptor and its overexpression can suppress *PP2Cs*, which releases *SNRK2s* from the inhibition of *PP2Cs* to subsequently activate its downstream target (ABRE-binding factor) [17,18]. 

Furthermore, ETH is involved not only with an array of physiological and developmental processes (from the regulation of organ growth to inducing fruit ripening), but also multiple stress responses [19]. Also, CK, SA, JA, and BR mediate stress adaptation responses in plants. For example, CK regulates cell proliferation, differentiation, leaf aging, and leaf complexity. JA and SA are often considered as resistant to related endogenous hormones in that they have key advantages for plant responses to various stresses. BR is a sterol compound that can regulate plant seed germination, flowering, senescence, tropism, photosynthesis, and stress resistance, which is closely related to other signaling molecules [20]. 

*Opisthopappus* genus is a perennial herb that general grows on the cliffs of the Taihang Mountains in China. Within *Opisthopappus* genus are two species (*Opisthopappus taihangensis* (*O. taihangensis*) and *Opisthopappus longilobus* (*O. longilobus*)). As a typical cliff plant both of these species exhibit good cold and drought resistance and are considered an important wild germplasm of Asteraceae [21,22,23]. 

When we inadvertently planted *O. taihangensis* and *O. longilobus* plants into the salt alkaline soil of our university, it was observed that both species grew well and exhibited good salt resistance; thus, we pondered what the underlying mechanism might be. Consequently, salt stress treatments for *O. taihangensis* and *O. longilobus* were conducted in the laboratory. Firstly, the concentration gradients of mixed salt solutions were set to 0, 100, 300, 500, and 700 mM. The chlorophyll, peroxidase (POD), catalase (CAT), superoxide dismutase (SOD), soluble protein (SP), and malondialdehyde (MDA) contents were then measured following the treatments. The results revealed that a salt concentration of 500 mM was a critical value for *O. taihangensis* and *O. longilobus*. Thus, we selected 500 mM as the salt concentration for the time gradient treatments of *O. taihangensis* and *O. longilobus*. 

After the treatments, the transcriptomic data of *O. taihangensis* and *O. longilobus* under salt stress with different time gradients was sequenced. Based on the outcomes, we aimed to address the following issues: (1) determine the expression trends of differentially expressed genes (DEGs) related to endogenous hormones under salt stress; (2) analyze the biosynthesis and signal transduction pathways of the endogenous hormones under salt stress; (3) explore the similarities and differences of the endogenous hormone responses to salt stress between *O. taihangensis* and *O. longilobus*. The results would provide a reference for exploring the salt tolerance mechanisms of *O. taihangensis* and *O. longilobus*, while laying a foundation for the study of the responses of endogenous hormones in other cliff plants.

## 2. Results

### 2.1. DEGs Related to Endogenous Opisthopappus Hormones

A total of 152.43 Gb of clean data was obtained after sequencing, with 77.06 Gb of clean data for *O. taihangensis* and 73.37 Gb of clean data for *O. longilobuss*. For each sample, >96.57% of the bases had scores of Q30 or above, which indicated that the sequencing results could be used for subsequent analysis (Table A1).

Based on the above results, the DEGs correlated with endogenous hormone biosynthesis and signal transduction pathways were screened. There were 239 DEGs between the *O. taihangensis* and *O. longilobus* obtained between 0 h vs. 6 h. Furthermore, 239 DEGs and 239 DEGs between 6 h vs. 24 h and 24 h vs. 48 h, respectively, were also obtained (Figure 1A). Thus, these DEGs were considered as common (C-DEGs).

The expression of 239 C-DEGs in *O. taihangensis* presented four change trends under different time gradient treatments. These change trends included Profile1, Profile 35, Profile 47, and Profile 48, respectively, in which Profile 47 was significantly upregulated and Profile 1 was significantly downregulated (Figure 1B–E). 

For *O. longilobus* the expression of 239 C-DEGs also had four change trends, which were Profile 44, Profile 45, Profile 46, and Profile 47, respectively. Among these four profiles, only Profile 47 was upregulated (Figure 1F–I). 

According to the above, the *O. taihangensis* genes in Profile 1 and Profile 47, and the *O. longilobus* genes in Profile 47 might play critical roles in the responses to salt stress.

### 2.2. KEGG of DEGs

KEGG revealed that the DEGs were primarily enriched in 10 pathways. Nine pathways were essentially same between *O. taihangensis* and *O. longilobus* (signal transduction, protein kinases, plant hormone signal transduction, metabolism of terpenoids and polyketides, MAPK signaling pathway–plant, environmental information processing, brassinosteroid biosynthesis, carotenoid biosynthesis, and alpha-Linolenic acid metabolism). Among these similar pathways, plant hormone signal transduction was the most significant between *O. taihangensis* and *O. longilobus* (Figure 2).

The remaining pathway (biosynthesis of various secondary metabolites—part 3) uniquely occurred in *O. taihangensis*, while linoleic acid metabolism was unique in *O. longilobus*. However, the significance of both these pathways was low.

Accordingly, the DEGs annotated in the plant hormone signal transduction and biosynthetic pathways of *O. taihangensis* and *O. longilobus* were further analyzed.

### 2.3. Biosynthesis and Signal Transduction Pathways of Endogenous Hormones

A total of 155 C-DEGs were discovered in the plant hormone signal transduction pathway (Figure 3A). This pathway mainly involved the signal transduction of AUX, CK, GA, ABA, ETH, BR, JA, and SA.

#### 2.3.1. AUX and GA Biosynthesis and Signal Transduction

Within the AUX biosynthesis pathway 13 C-DEGs were involved, having two L-tryptophan–pyruvate aminotransferase genes (*TAA1s*), two indole-3-pyruvate monooxygenase genes (*YUCCAs*), two aromatic-L-amino-acid/L-tryptophan decarboxylase genes (*DDCs*), four aldehyde dehydrogenase genes (*ALDHs*), and three amidase genes (*amiEs*). The expressions of these genes were basically the same between *O. taihangensis* and *O. longilobus*.

Meanwhile, a total of thrity-nine C-DEGs were involved in AUX signal transduction, including three auxin influx carrier genes (*AUX1s*), three transport inhibitor response 1 genes (*TIR1s*), nine auxin-responsive protein IAA genes (*IAAs*), seven auxin response factor genes (*ARFs*), six auxin-responsive GH3 gene family genes (*GH3s*), and eleven SAUR family protein genes (*SAURs*). 

The expression levels of most *AUX1s* and *IAAs* presented peaks at 0 h in *O. longilobus*, while they appeared as low points for *O. taihangensis*. At 48 h, the expression of *TIR1s* in *O. taihangensis* was significantly higher than that in *O. longilobus*, while the expression of *GH3s* exhibited a similar trend, reaching its peak at 48 h in *O. taihangensis*. Nevertheless, the expression of *ARFs* was essentially identical for *O. taihangensis* and *O. longilobus*. Interestingly, the expression of six *SAURs* from *O. longilobus*, and five *SAURs* from *O. taihangensis* were downregulated at all time points under the salt stress treatments (Figure 3B). 

Five C-DEGs encoding for gibberellin 3beta-dioxygenase genes (*GA3ox*) were found to be involved in GA synthesis in the diterpenoid biosynthesis pathway. Notably, some of these genes exhibited low expression levels at 6, 24, and 48 h in *O. taihangensis* and *O. longilobus*. 

Further, 21 C-DEGs associated with GA signal transduction were detected, which included three gibberellin receptor GID1 genes (*GID1s*), 10 DELLA protein genes (*DELLAs*), three F-box protein GID2 genes (*GID2s*), one phytochrome-interacting factor 4 gene (*PIF4*), and four phytochrome-interacting factor 3 genes (*PIF3s*). 

*GID1s* was upregulated at 0, 6, 24, and 48 h in *O. taihangensis*, while it was downregulated at 48 h in *O. longilobus*. Most *DELLAs* exhibited upregulation at 48 h and reached their peak expression levels in *O. taihangensis*. A few *DELLAs* exhibited upregulation at 48 h in *O. longilobus*. Moreover, a significant difference was observed in the expression levels of *GlD2s* between *O. longilobus* and *O. taihangensis* at 48 h, with a higher expression in *O. longilobus*. Similarly, the expression levels of *PIF3s* at 48 h were notably lower in *O. longilobus* compared with *O. taihangensis* (Figure 3C).

#### 2.3.2. JA and BR Biosynthesis and Signal Transduction

There were a total of 26 C-DEGs in the JA biosynthesis pathway, including 10 lipoxygenase genes (*LOX2Ss*), 1 hydroperoxide dehydratase gene (*AOS*), 1 allene oxide cyclase gene (*AOC*), 3 12-oxophytodienoic acid reductase genes (*OPRs*), 3 OPC-8:0 CoA ligase 1 genes (*OPCL1s*), 4 acyl-CoA oxidase genes (*ACXs*), and 4 enoyl-CoA hydratase/3-hydroxyacyl-CoA dehydrogenase genes (*MFP2s*). Among the above, the expressions of *ACXs* and *MFP2s* in *O. taihangensis* and *O. longilobus* were upregulated at 6, 24, and 48 h. 

A total of 21 C-DEGs were observed in JA signal transduction, namely 1 jasmonic acid-amino synthetase gene (*JAR1_4_6s*), 2 coronatine-insensitive protein 1 genes (*COI-1s*), 8 jasmonate ZIM domain-containing protein genes (*JAZs)*, and 10 transcription factor MYC2 genes (*MYC2s*). 

The *JAR1_4_6s* expression was upregulated at 0 and 48 h in *O. longilobus*, while upregulated at 0 h and downregulated at 48 h in *O. taihangensis*. The expression of *COI-1s* was upregulated at 6, 24, and 48 h in *O. longilobus*, while that of *COI-1s* in *O. taihangensis* was significantly lower than in *O. longilobus*. For *O. taihangensis* and *O. longilobus* the expression of *JAZs* was upregulated at 6 h. *MYC2s* reached its peak at 6 h, and two *MYC2s* were upregulated at 48 h in *O. longilobus*. At 6 and 48 h, the expression of *MYC2s* in *O. taihangensis* was significantly lower than that in *O. longilobus* (Figure 3D).

There were seven C-DEGs involved in BR biosynthesis pathway, including one steroid 22S-hydroxylase gene (*CYP90B*), one 3-epi-6-deoxocathasterone 23-monooxygenase gene (*CYP90D1*), two 3beta,22 alpha-dihydroxysteroid 3-dehydrogenase genes (*CYP90A1s*), two typhasterol/6-deoxotyphasterol 2alpha-hydroxylase genes (*CYP92A6s*), and one brassinosteroid 6-oxygenase gene (*CYP85A1*). 

The expression of *CYP90D1* was upregulated at 6 and 24 h in *O. longilobus*, as well as at 24 and 48 h in *O. taihangensis*. In *O. taihangensis*, the expression of *CYP90A1s* was upregulated at 6, 24, and 48 h. The expression of *CYP92A6s* was upregulated at 6 and 48 h in *O. longilobus*, while it was upregulated at 6 h in *O. taihangensis*. 

In addition, 15 C-DEGs involved in BR signal transduction were found, namely 2 brassinosteroid-insensitive 1-associated receptor kinase 1 genes (*BAK1s*), 4 protein brassinosteroid-insensitive 1 genes (*BRI1s*), five BR-signaling kinase genes (*BSKs*), 3 xyloglucan:xyloglucosyl transferase TCH4 genes (*TCH4s*), and 1 cyclin D3 gene (*CYCD3*). Four *BRI1s* from *O. longilobus* and one *BRI1* from *O. taihangens* were upregulated after 48 h. The expression of *BSKs* at 6, 24, and 48 h in *O. taihangensis* was higher than that in *O. longilobus*. Most *TCH4s* and *CYCD3* reached their expression peaks at 0 h in *O. longilobus*; however, they were downregulated in *O. taihangensis* (Figure 3E).

#### 2.3.3. ABA and ETH Biosynthesis and Signal Transduction

A total of 20 C-DEGs were found to be involved in the ABA biosynthesis pathway, including 1 beta-carotene 3-hydroxylase gene (*crtZ*), 1 beta-ring hydroxylase gene (*LUT5*), 3 zeaxanthin epoxidase genes (*ZEPs*), 3 9-cis-epoxycarotenoid dioxygenase genes (*NCEDs*), 11 xanthoxin dehydrogenase genes (*ABA2s*), and 1 abscisic-aldehyde oxidase gene (*AAO3*), among which 1 *ZEP* in *O. taihangensis* was significantly higher than that of *O. longilobus* at 48 h. 

There were 24 C-DEGs involved in ABA signal transduction, namely 1 abscisic acid receptor PYR/PYL family gene (*PYR/PYL*), 9 protein phosphatase 2C genes (*PP2Cs*), 9 serine/threonine-protein kinase SRK2 genes (*SNRK2s*), and 5 ABA responsive element binding factor genes (*ABFs*). In *O. longilobus*, the expression of *PYL* was upregulated at 0 h, which was downregulated in *O. taihangensis*. Most *PP2Cs* exhibited upregulated expression at 6 and 24 h and reached their peaks at 24 h in *O. longilobus*. In contrast, the expression of *PP2Cs* peaked in *O. taihangensis* at 48 h. Most of the *SNRK2* genes peaked in both species at 6 h. Meanwhile, the downstream *ABFs* of *PP2C* genes reached their peaks at 24 h and began to downregulate at 48 h in *O. longilobus*. For *O. taihangensis*, *ABFs* continued to increase and reached their peak at 48 h (Figure 4A).

A total of 13 C-DEGs were involved in ETH biosynthesis pathway, having 7 S-adenosylmethionine synthetase genes (*metKs*), 3 1-aminocyclopropane-1-carboxylate synthase 1/2/6 genes (*ACS1_2_6s*), 1 1-aminocyclopropane-1-carboxylate synthase gene (*ACS*), and 2 aminocyclopropanecarboxylate oxidase genes (*E1.14.17.4s*). The expression of *metKs* was upregulated at 6 h in *O. longilobus*, whereas *E1.14.17.4s* was upregulated at 6, 24, and 48 h in *O. longilobus*.

Meanwhile, 13 C-DEGs were involved in ETH signal transduction, including 4 ethylene receptor genes (*ETRs*), 1 serine/threonine-protein kinase CTR1 gene (*CTR1*), 1 mitogen-activated protein kinase kinase 4/5 gene (*MKK4_5*), 2 ethylene-insensitive protein 2 genes (*EIN2s*), 2 ethylene-insensitive protein 3 genes (*EIN3s*), 2 EIN3-binding F-box protein genes (*EBF1_2s*), and 1 ethylene-responsive transcription factor 1 gene (*ERF1*). Among them, the expression of *ETRs* was upregulated at 48 h in both *O. taihangensis* and *O. longilobus*, as were *CTR1*, *MKK4_5*, and *EIN2s*. The expression of one *EIN3* was highest at 0 h in *O. longilobus*, while the expression of another *EIN3* was the highest at 48 h in *O. taihangensis*. *EBF1_2s* were upregulated at 6 h in *O. longilobus*, whereas only one was upregulated at 48 h in *O. taihangensis*. The expression of *ERF1* was downregulated at 48 h for *O. longilobus*; however, it was upregulated at 24 and 48 h in *O. taihangensis* (Figure 4B).

#### 2.3.4. CK and SA Biosynthesis and Signal Transduction

Unfortunately, no C-DEGs were observed in the CK and SA biosynthesis pathways. However, 16 C-DEGs were found to be involved in CK signal transduction, including 2 arabidopsis histidine kinase 2/3/4 genes (*AHK2_3_4s*), 12 two-component response regulator ARR-B family genes (*ARR-Bs*), and 2 two-component response regulator ARR-A family genes (*ARR-As*). Among them, the expression of *ARR-Bs* was upregulated at 48 h in *O. taihangensis*, while it was downregulated at the same time in *O. longilobus* (Figure 4C). 

A total of six C-DEGs were involved in SA signal transduction, including one regulatory protein NPR1 gene (*NPR1*), two transcription factor TGA genes (*TGAs*), and three pathogenesis-related protein 1 genes (*PR-1s*). Meanwhile the expression of *PR-1s* was upregulated at 48 h in *O. longilobus* (Figure 4D).

### 2.4. MAPK Signaling Pathway

The MAPK signaling pathway in this study was also significantly enriched, and 50 C-DEGs were found, which were consistent with the above signaling pathways of ETH, ABA, JA, BR, and SA.

A total of 16 C-DEGs were identified in the ETH signaling pathway, including 3 *ACS1-2-6s*, 4 *ETRs*, 1 *CTR1*, 1 *MKK4_5*, 2 *EIN2s*, 2 *EIN3s*, 2 *EBF1_2s*, and 1 *ERF1*. 

A total of 10 *MYC2s* were found in the JA signaling pathway. The expressions of *CTR1*, *MKK4_5*, and *EIN2s* were upregulated at 48 h in both in *O. taihangensis* and *O. longilobus*. *ERF1* in *O. taihangensis* had a significantly higher expression level than that in *O. longilobus* at 24 and 48 h, respectively. 

Nineteen C-DEGs were involved in the ABA signaling pathway, including one *PYL*, nine *PP2Cs*, and nine *SNRK2s*. Among them, *PP2Cs* were upregulated in *O. taihangensis* and *O. longilobus* at both 6 and 24 h. However, it was noteworthy that *PP2Cs* were continuously upregulated in *O. taihangensis* and downregulated in *O. longilobus* at 48 h. 

Two *BAK1s* were found in the BR signaling pathway, while three *PR1s* were observed in the SA signaling pathway (Figure 5).

### 2.5. Interactive Network Analysis

A total of 65 protein–protein interaction (PPI) IDs were uploaded from the Sting online database. These proteins were primarily involved in the biosynthesis of plant hormones and signal transduction pathways (Figure 6). They revealed that 29 proteins formed an interactive network. Furthermore, the highest interactions of proteins included BRI1, AUX1, EIN2, BAK1, EIN3, ABA2, and GA3OX1, respectively.

### 2.6. qRT-PCR Quantitative Verification 

To verify the accuracy of the sequencing data, three C-DEGs were selected for expression verification by qRT-PCR. These genes were involved in the biosynthesis and signal transduction of AUX, ETH, and ABA. Based on the qRT-PCR, the relative expression levels of three C-DEGs revealed a consistent trend with the above results, which confirmed the accuracy of the sequencing data (Figure 7).

## 3. Discussion

Endogenous hormones can lead to rapid and sustained changes in gene regulation when plants respond to salt stress and play important roles in plant growth and development, as well as the mitigation of salt stress [5]. This study identified 239 C-DEG-related endogenous hormones that were significantly enriched in AUX, GA, JA, BR, EHT, ABA, and MAPK signal transduction pathways in *O. taihangensis* and *O. longilobus* under salt stress. A total of 29 proteins having the highest interactions were found in the PPI network and involved in these pathways, which implied crosstalk between the above hormones.

AUX regulates plant morphogenesis under salt stress, including vegetative growth and reproduction [7]. AUXIN1/LIKE-AUX1 (AUX1/LAX) family members are the major auxin influx carriers implicated in regulating key processes including root and lateral root development, root gravitropism, root hair development, vascular patterning, seed germination, apical hook formation, leaf morphogenesis, phyllotactic patterning, female gametophyte development, and embryo development [24]. Recently, AUX1 (Auxin transporter protein 1, encoded by *AUX1* gene) was also implicated in the regulation of plant responses to abiotic stresses [25]. Our results showed that AUX1 possessed the highest PPI interactivity (Figure 6). Moreover, there was no difference in the *AUX1* expression between the two species under salt stress (Figure 3B).

Furthermore, six *SAURs* of AUX from *O. longilobus*, and five *SAURs* from *O. taihangensis* were inhibited at all time points under the salt stress treatments (Figure 3B). Auxin, *TIR1*, and *ABF* combined to form a complex that led to the ubiquitination and degradation of *AUX/IAA*, inhibited the functions of auxin response factors (*ARFs*), and negatively affected the expression of downstream transcription factor small auxin-upregulated RNAs (*SAUR*) [26]. *SAURs* can regulate the division and expansion of plant cells related to plant morphogenesis. The overexpression of *SAUR* in *A. thaliana* can result in cell elongation [27], whereas in *Benincasahispida* it has been found to be associated with longer floral organs and wavy stems [28]. Thus, the inhibition of *SAURs* in *O. taihangensis* and *O. longilobus* might delay morphogenesis under salt stress, serving as a survival strategy for these two species.

GA3ox is encoded by *GA3ox* gene, which is a key enzyme in the final step of the GA biosynthesis pathway [29]. The reduced accumulation of GA can slow growth and increase the contents of soluble sugar and chlorophyll, which can enhance the salt tolerance of plants [30]. It was found that the expressions of GA biosynthesis genes were downregulated in *O. taihangensis* and *O. longilobus* (Figure 3C). This suggested that under salt stress the reduction of GA biosynthesis retarded the growth of *O. taihangensis* and *O. longilobus*. 

*DELLA* is a negative regulator of the GA signal transduction pathway [29], and studies indicated that its accumulation could stimulate defenses against biotic and abiotic stressors, while repressing cell division and expansion in angiosperms [31]. The expression of *DELLA* was upregulated in both *O. taihangensis* and *O. longilobus*; however, its expression in *O. taihangensis* was significantly higher than that in *O. longilobus* at 48 h under salt stress (Figure 3C). The higher expression of GA negative regulatory factors in *O. taihangensis* indicated that this species might augment salt resistance by decelerating its growth.

In the JA biosynthesis pathway, the expression of both *ACX* and *MFP2* were upregulated in *O. taihangensis* and *O. longilobus*. The increased JA biosynthesis in *Arabidopsis* and wheat improved their salt tolerance, while a reduction in JA production or accumulation translated to high salt sensitivity in tomatoes and rice [32,33]. JA-mediated growth inhibition may be an important adaptive strategy in saline environments. 

Once JA is sensed by the *COI1* receptor, it forms an *SCFCOI1-E3* ligase complex with *SKP1* and *CULLIN1* [34]. This complex mediates the degradation of *JAZ* by the 26S proteasome and releases the inhibition of JA response genes (e.g., *MYC*), which then activates JA signaling [35]. JAZ proteins are the negative regulators of JA signaling, which play a critical role in the responses of plants to salt stress. Several *JAZ* homologous genes were observed to be upregulated under NaCl treatments in cotton, *Arabidopsis* roots, tomato, and wheat [35,36,37]. Moreover, the overexpression of *OsJAZ9* in rice results in a higher tolerance to salt stress [38]. In this study, the expression of *JAZ* was upregulated at 6 h in *O. taihangensis* and *O. longilobus* (Figure 3D). 

Generally, BR signaling initially begins with *BRI1*, which is a cell surface receptor kinase [39]. The extracellular BRI1 domain recognizes BRs that leads to heteromerization with *BAK1*, which is a member of the somatic embryogenesis receptor kinase (*SERK*) family of proteins. Subsequently, intracellular *BRI1* and *BAK1* kinase domains induce transphosphorylation, which triggers a downstream signaling cascade that eventually leads to the expression or suppression of downstream *BRI1* and *BAK1* gene overexpression, which are typically correlated with a tolerance for high salt stress [40]. Our results revealed that *BAK1s* were upregulated in *O. taihangensis* and *O. longilobus* after 48 h under the salt stress treatments. Four *BRI1s* from *O. longilobus* and one *BRI1* from *O. taihangens* were also upregulated after 48 h. It was concluded that BR signal transduction may play a certain role in the salt tolerance of *O. taihangensis* and *O. longilobus* (Figure 3E).

ABA is defined as a stress hormone due to its rapid accumulation in response to stress, which can mediate multiple stress responses to assist with the survival of plants. *ZEP* is a key regulatory gene in the ABA biosynthesis pathway of plants [41]. Previous studies have indicated that *ZEP* overexpressing types in *Arabidopsis thaliana* exhibited more vigorous growth under high salt and drought treatments than wild types [42]. Simultaneously, one *ZEP* in *O. taihangensis* was significantly higher than that of *O. longilobus* at 48 h. In addition to *ABF*, it is considered to be a core factor involved in the ABA signaling pathway. *ABFs/AREBs* regulate stomatal closure and leaf senescence to response to abiotic stresses, such as salt, drought, heat and cold [43,44,45,46]. In this study, the expression of *ABF* was continuously upregulated in *O. taihangensis* at 6, 24, and 48 h, while it was upregulated only at 6 and 24 h in *O. longilobus* (Figure 4A). Additionally, *ABFs* play a certain role in *O. taihangensis* when faced to drought stress [23]. These results indicated that *O. taihangensis* and *O. longilobus* responded to abiotic stress via the accumulation of ABA. Further, *O. taihangensis* exhibited a higher salt tolerance than *O. longilobus* due to its capacity to accumulate additional ABA.

SNRK2 is encoded by the *SNRK2* gene and plays essential roles in the abiotic stress responses of plants as a positive global regulator of abscisic acid signaling. The overexpression of SNRK2 in *Arabidopsis thaliana* maintained higher chlorophyll levels and longer root systems under salt stress, and its survival rate was significantly higher than that of the wild type [47]. In this study, most *SNRK2* genes peaked at 6 h for both species (Figure 4A). This suggested that the reduction in ABA signal transduction under salt stress slowed the growth of *O. taihangensis* and *O. longilobus*.

The responses of ETH to salt stress varied significantly between various plants. For instance, ETH signal transduction was confirmed to promote salt tolerance in *Arabidopsis*; however, in rice the ETH signals negatively regulated salt tolerance. Several studies revealed that *ETR* mutants could enhance salt tolerance [48]. In this study, the expression of *ETRs* was upregulated at 48 h for *O. taihangensis* and *O. longilobus* (Figure 4B). The EIN2 nuclear protein, which has higher interactivities with other proteins (Figure 6), is a core component of the ETH signal transduction pathway in plants, which plays an important role in mediating crosslinks between hormone response pathways, such as ABA. *EIN2* is required to induce developmental arrest during seed germination, seedling establishment, as well as subsequent vegetative growth, which then enables plants to survive and grow under adverse environmental conditions [49]. In *O. taihangensis* and *O. longilobus*, the expression of *EIN2s* was upregulated at 6, 24, and 48 h. Consequently, ETH signal transduction likely regulated salt tolerance in *O. taihangensis* and *O. longilobus*.

In PPI, ERF1 had fewer correlations with other genes; however, there was an interaction between EIN2 and ERF1 (Figure 6). *ERF1* was the downstream gene of *EIN2* in ETH signal transduction, which revealed significant differences in *O. taihangensis* and *O. longilobus*. Under salt stress the expression of *ERF1* in *O. taihangensis* was significantly higher than that in *O. longilobus* at 24 and 48 h (Figure 4B). Many *Arabidopsis ERFs* can regulate genes under abiotic stresses. *ERF1* and *ESEs* (ethylene-and salt-inducible ERF genes) in the *ERF-IX* group positively regulated plant salinity tolerance by promoting salt responsive gene expression [50]. Thus, it was inferred that higher ETH signal transduction in *O. taihangensis* conferred greater tolerance than in *O. longilobus* under long-term salt stress.

If MAPK signaling pathway is activated, it initiates the phosphorylation of downstream signaling targets and responds to diverse changes in the extracellular or intracellular environment. Subsequently the phosphorylated targets exert regulatory control over cellular, organ, or organismal metabolism [51]. Additionally, MAPK protein kinases exert influence over diverse intracellular responses and functions encompassing inflammation, cell-cycle regulation, differentiation, development, senescence, and death [52]. Based on our results, 50 C-DEGs were significantly enriched in the MAPK signaling pathway in *O. taihangensis* and *O. longilobus* under salt stress. 

Recently, the crosstalk mechanisms between MAPK cascades and endogenous hormones, including AUX, ETH, ABA, JA, SA, and BR, were identified in plants, where the overexpression of *ERFs* (ETH signal transduction pathway) enhanced the resistance of salinity by activating the MAPK signaling cascade [50]. In this study, DEG-related ETH, ABA, SA, BR, and JA were identified in the MAPK signal pathway under salt stress. Further, the expression of *ERF1* (ETH gene) in *O. taihangensis* was significantly higher than that in *O. longilobus* at 24 and 48 h. This implied that ETH-related genes enriched with MAPK signaling were engaged in resistance to salt stress (Figure 5).

In this work, comparative transcriptome analysis was performed on *O. taihangensis* and *O. longilobus* to explore the potential mechanisms behind their responses to salt stress. The results revealed that the two species responded to salt stress primarily through crosstalk between GA, JA biosynthesis, and signal transduction, ABA, ETH, AUX, and BR signal transduction pathways. Moreover, *O. taihangensis* exhibited a relatively higher salt tolerance than *O. longilobus* with the higher expression of some genes in the above pathways, such as *ZEP*, *DELLA*, *ABF*, and *ERF1*, which facilitated its good adaptation to the cliff environment of the Taihang Mountains (Figure 8).

## 4. Materials and Methods

### 4.1. Plant Materials and Salt Treatments

The *O. taihangensis* and *O. longilobus* seeds were collected in 2021 from the common garden of Shanxi Normal University (111°30′ W, 36°06′ N), in Shanxi Province, China. Healthy seeds were selected and germinated in Petri dishes in the laboratory at room temperature. After germination, the seeds with consistent growth were selected and transplanted into a seedling tray with a sterilized substrate for culturing. Three weeks later, the seedlings with strong growth were again transplanted into plastic pots and continued to grow at room temperature.

After six weeks of culturing, the healthy and consistently growing seedlings were selected for the salt stress treatments. Subsequently, fresh leaves from the same parts of each sampled individual were collected after 0, 6, 24, and 48 h of treatment at 500 mM salt concentrations. Three individuals sustained with distilled water were used as the control group, and each treatment was repeated three times. 

Following the treatments, a total of 24 samples were frozen in liquid nitrogen, and transferred to Lc-Bio Technologies (Hangzhou, China) Co., Ltd. for transcriptome sequencing.

### 4.2. DEGs Related to Endogenous Hormones

The DEGs related to the endogenous hormones were initially derived and identified via the transcriptome data. Subsequently, the DEGs related to the biosynthesis and signal transduction pathways of AUX, CK, GA, ABA, ETH, BR, JA, and SA were screened from 0 h vs. 6 h. vs. 24 h vs. 48 h under the salt stress treatments. A Venn diagram was then employed to screen the common DEGs (C-DEGs) related to the endogenous hormones between *O. taihangensis* and *O. longilobus*.

Simultaneously, a STEM tool (Lianchuan Biological Cloud Platform, https://www.omicstudio.cn/index, accessed on 26 June 2023) was used to analyze the expression trends of the identified C-DEGs.

### 4.3. Enrichment Analysis

The Kyoto Encyclopedia of Genes and Genomes (KEGG) enrichment analyses of identified DEGs related to the endogenous hormones were performed using the TB-tools. The hypergeometric distribution principle was adopted in the enrichment analysis, and the identified DEG sets were employed through the analysis of significant expression differences. Subsequently, these DEGs were annotated to the KEGG database (https://www.genome.jp/kegg/pathway.html (accessed on 1 July 2023). The background genes were established; these consisted of all genes subjected to the significant difference analysis and annotated to the KEGG database. A KEGG enrichment map was then generated using the Lianchuan Biological Cloud Platform. The enriched pathways were classified and annotated, and those that were significantly enriched were selected for further analysis.

### 4.4. C-DEGs Analysis

Initially, the expression trends of the C-DEGs were identified by annotating to the endogenous hormone signaling pathway. The Fragments Per Kilo Base Per Million Mapped Reads (FPKM) values of the C-DEGs within the signaling pathways of AUX, CK, GA, ABA, ETH, BR, JA, and SA were evaluated to discern their expression patterns. Additionally, the C-DEGs linked to the hormone biosynthesis process were scrutinized. Since the MAPK signaling pathway had the capacity to modulate plant tolerance to salt stress by crossing other signaling pathways, the DEGs were also annotated in accordance with the MAPK signaling pathway.

### 4.5. PPI Network of DEGs

The network analysis of PPIs was performed to uncover plausible interactions between proteins with candidate genes involved in the endogenous hormone biosynthesis and signal transduction of AUX, CK, GA, ABA, ETH, BR, JA, and SA. The PPIs were analyzed using String (https://cn.string-db.org/) (accessed on 10 July 2023), *Arabidopsis thaliana* was used as the reference data, and the network was visualized using Cytoscape (3.9.0).

### 4.6. Quantitative Real-Time PCR (qRT-PCR)

To verify the credibility of the DEG trends in each signaling pathway, three C-DEGs genes (evm. TU. Chr8.9802, evm. TU. Chr5.12740, and evm. TU. Chr8.39) were randomly selected for quantitative qRT-PCR analysis. During the qRT-PCR, three biological and three technical replicates were adopted for each gene.

Using the Prime Script™ RT Reagent Kit (Takara, Japan), the RNA of the samples was initially reversed to cDNA, after which the synthesized cDNA was used as a template for quantitative qRT-PCR. 

The qRT-PCR was performed under the following conditions: 95 °C for 3 min, followed by 40 cycles of 95 °C for 5 s, and at 60 °C for 20 s. For the PCR, actin (evm. TU. Chr8.13443) was selected as the internal control gene. Finally, the expression levels of the selected genes were calculated using the 2^−ΔΔCt^ method. 

The primers are presented in Table A2. Significant differences between the DEGs were elucidated using one-way analysis of variance (ANOVA). **** represents a significant difference of *p* < 0.0001; *** represents a significant difference of *p* < 0.001; ** represents a significant difference of *p* < 0.01; * represents a significant difference of *p* < 0.05. GraphPad Prism 9.5.1 and Microsoft Excel 2009 were used for data analyses.

## Figures and Tables

**Figure 1 plants-13-00557-f001:**
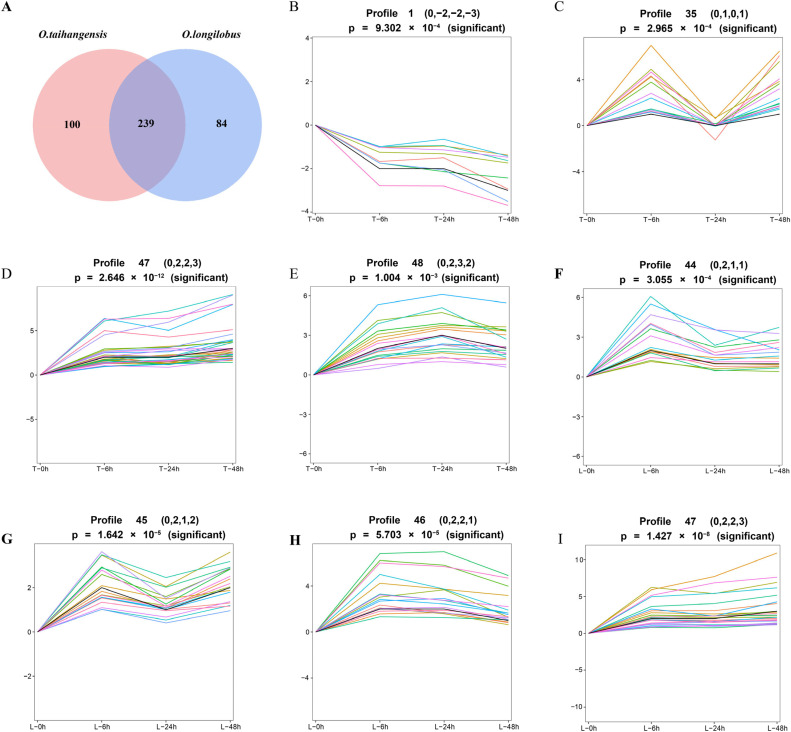
DEGs of *Opisthopappus taihangensis* (*O. taihangensis*) and *Opisthopappus longilobus* (*O. longilobus*) at different times under salt treatments. (**A**) Venn diagram of DEGs of *O. taihangensis* and *O. longilobus* for 0 h vs. 6 h. vs. 24 h vs. 48 h; (**B**–**E**) expression trend of 239 C-DEGs related to endogenous hormones in *O. taihangensis*; (**F**–**I**) expression trend analysis of 239 C-DEGs related to endogenous hormones in *O. longilobus*. Note: Horizontal axis represents the salt treatment time (0, 6, 24, and 48 h). T represents *O. taihangensis*, L represents *O. longilobus*.

**Figure 2 plants-13-00557-f002:**
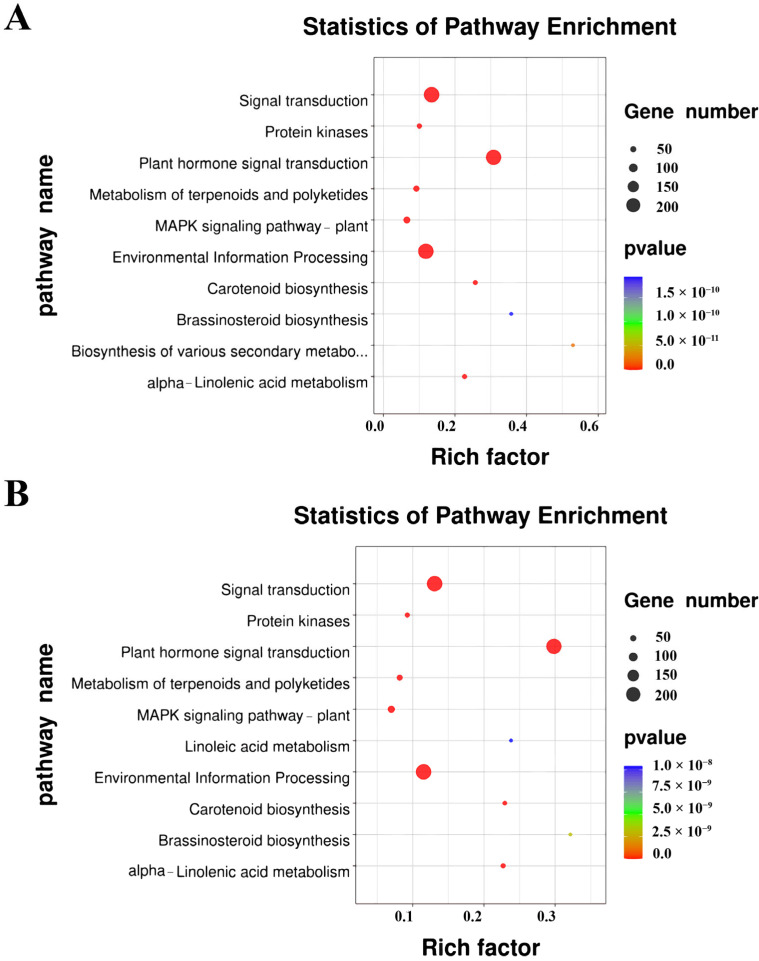
KEGG of DEGs (**A**) in *O. taihangensis*; (**B**) in *O. longilobus*.

**Figure 3 plants-13-00557-f003:**
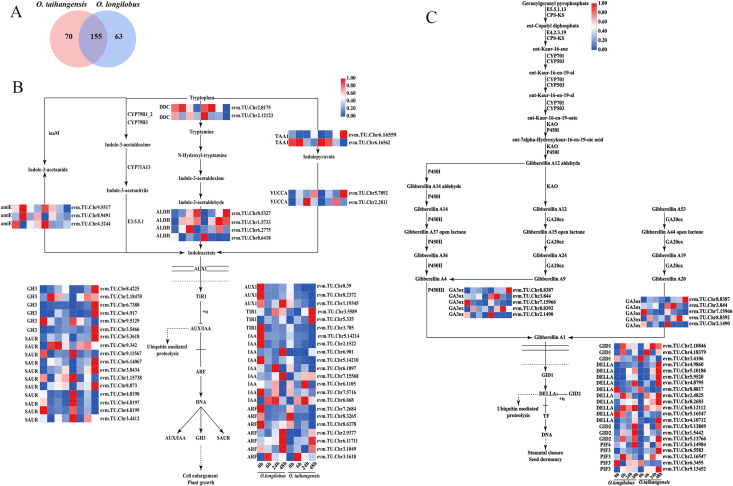
Different gene expression profiles of auxin, gibberellin, jasmonic acid, and brassinolide. (**A**) Venn diagram of DEGs of plant hormone signal transduction of *O. taihangensis* and *O. longilobus* in 0 h vs. 6 h. vs. 24 h vs. 48 h; (**B**) auxin; (**C**) gibberellin; (**D**) jasmonic acid; (**E**) brassinolide. Note: the red-blue schemes labelled on the right side of the heat map, and red to blue represent the expression levels from high to low.

**Figure 4 plants-13-00557-f004:**
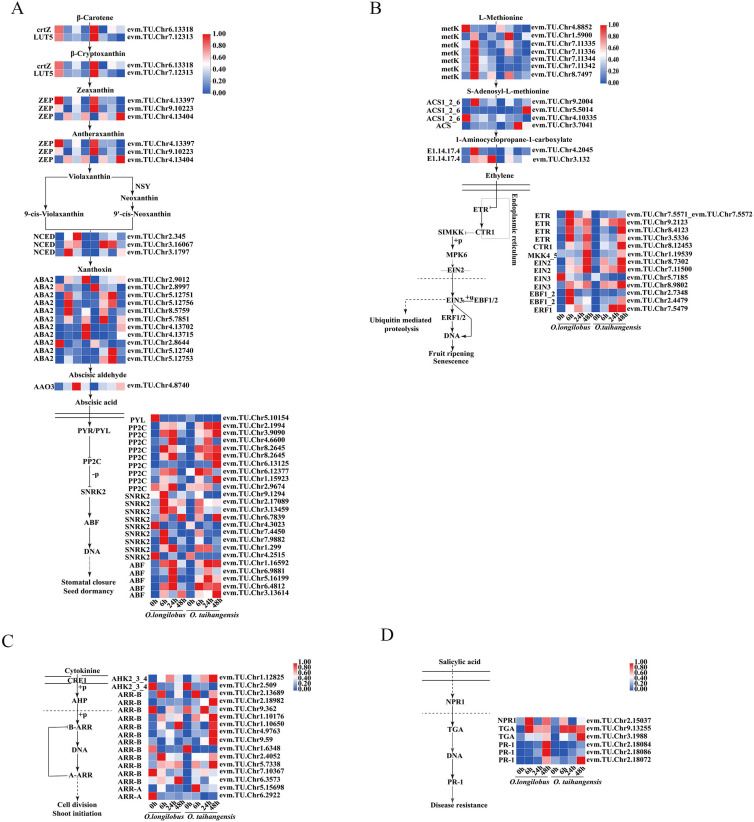
Differential gene expression profiles of abscisic acid, ethylene, cytokinin, and salicylic acid. (**A**) abscisic acid; (**B**) ethylene; (**C**) cytokinin; (**D**) salicylic acid. Note: the red-blue schemes are labelled on the right side of heat map, and red to blue represent the expression levels from high to low.

**Figure 5 plants-13-00557-f005:**
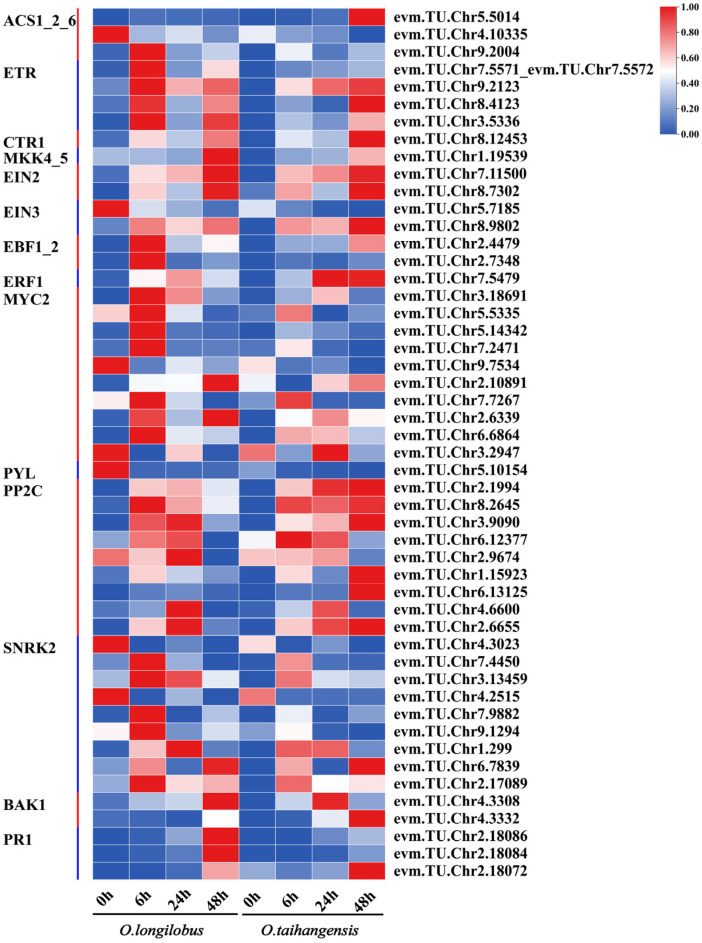
Heat maps of DEGs in the MAPK signaling pathway of *O. taihangensis* and *O. longilobus*. Note: Red-blue schemes are labelled on the right side of the heat map, and red to blue represents the expression levels from high to low.

**Figure 6 plants-13-00557-f006:**
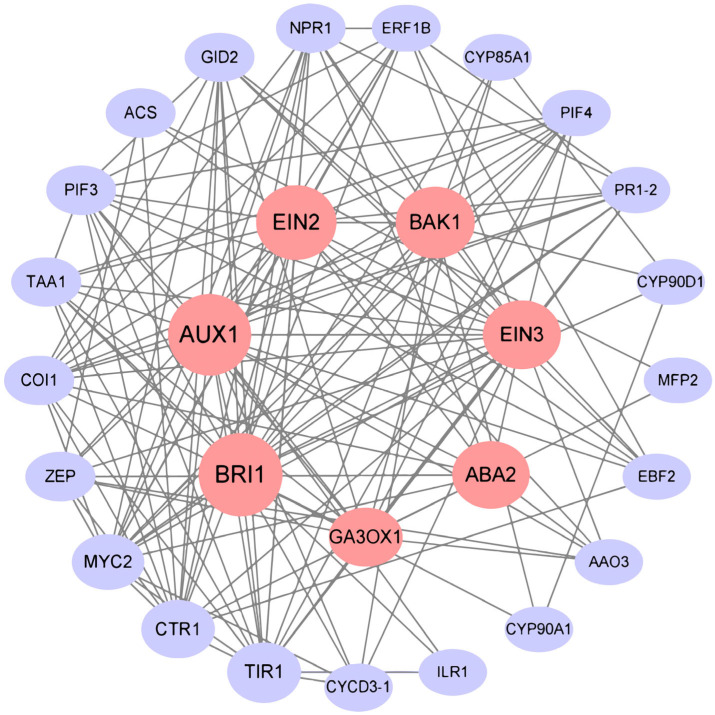
Protein–protein interaction (PPI) network in biosynthesis of plant hormones and signal transduction pathways. Red nodes represent the top seven highest interactions.

**Figure 7 plants-13-00557-f007:**
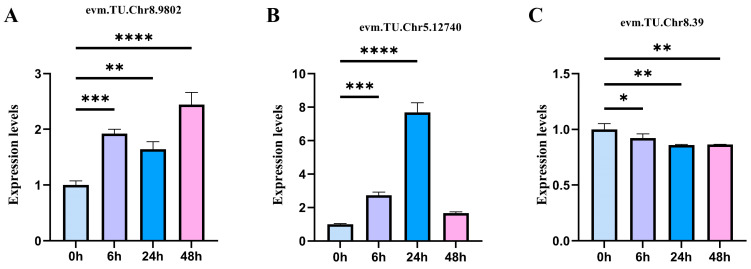
qRT-PCR verification of differentially expressed genes (DEGs). Relative gene expression levels under a 500 mM salt concentration for different time treatments (0, 6, 24, and 48 h). Vertical bars indicate the mean ± SD calculated from three replicates. Statistical comparisons (one-way analysis of variance (ANOVA) are presented for each variable (**** *p* < 0.0001 *** *p* < 0.001 ** *p* < 0.01 * *p* < 0.05). (**A**–**C**) *O. taihangensis*, (**D**–**F**) *O. longilobus*.

**Figure 8 plants-13-00557-f008:**
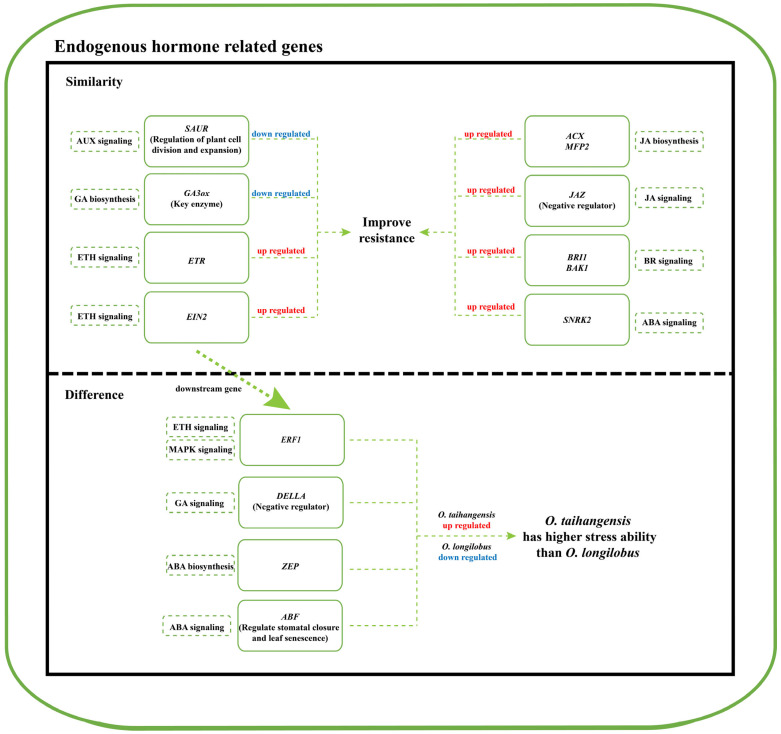
Summary of the responsive patterns of DEGs endogenous hormones between *O. taihangensis* and *O. longilobus* under salt stress.

## Data Availability

Data are contained within the article.

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
