# Peer review of "Potential Response Patterns of Endogenous Hormones in Cliff Species Opisthopappus taihangensis and Opisthopappus longilobus under Salt Stress"

_plants, 2024, doi:10.3390/plants13040557_

Round 1
Reviewer 1 Report
Comments and Suggestions for Authors
From this reviewer's perspective, the research approach, methods and inferences drawn are sound and compelling. My concerns primarily revolve around the rationale for your study of this pair of Opisthopappus species--why are they important to the broader plant community (vis-a-vis salt stress tolerance). Might their tolerance to drought and cold be related to their salt tolerance? That could be an extraordinarily rich point to bring out (assuming that this is a nontrivial association). In addition, I am unsure about the need to study cliff-dwelling plants--perhaps you are writing for an audience that is completely committed to this as an important value. However, this reviewer would argue that it is incumbent upon you as authors to "sell" the importance of studying such plants to the broader plant science research community. Other comments are included in the attached copy of your manuscript.

Sound--I only noted one tiny issue ("chlorophyl" rather than the correct spelling of "chlorophyll."). Just needs a careful check to make sure that all is good throughout (in terms of English usage).
Reviewer 2 Report
Comments and Suggestions for Authors
generally could be improved
Reviewer 3 Report
Comments and Suggestions for Authors
The authors of the manuscript propose as our main question the study of the patterns of potential responses in the contents of endogenous hormones in cliff species Opisthopappus taihangensis and Opisthopappus Longilobus under salt stress. The results of this research shed light on the possible adaptive mechanisms of O. taihangensis and O. longilobus under cliff environments, while laying the foundation for
the study of other cliff species in the Taihang Mountains.
Its objectives are clear, and they address questions of interest, such as (1) determining the expression trends of differentially expressed genes (DEGs) related to endogenous hormones under salt stress; (2) analyze the biosynthesis and signal transduction pathways of endogenous hormones under salt stress; (3) explore the similarities and differences of endogenous hormonal responses to salt stress between O. taihangensis and O. longilobus, so I consider the topic to be relevant, and it is a fairly complete work.
There is not much research related to these species, they are fundamentally directed at the study of molecular phylogeography, genetic diversity and micro satellites, which also makes it very original and relevant.
The methodology is complex and is very well described, there is an error in the NaCl concentration they use, it is not 500 mM/L, it is just 500 mM, I hope it is a typing error and not a conceptual error, also in the scientific names the species is always in lower case
The manuscript has no conclusions, at least not stated, but the discussion reports what the research reached and its importance.
The references are appropriate, 30% are from 2019 to date, which represents a low percentage, I think for the same reason that there is not much literature on the matter.
Regarding the figures and tables, we can comment that figures 3 and 4 have very small letters and that can be a problem when reading the article, the tables are relevant
I think it should be accepted after reviewing the comments.
Comments on the Quality of English LanguageIt is strongly recommended that the text be reviewed by an English speaker
